# Learning Binary Multi-Scale Games on Networks

**Sixie Yu**[1]      **P. Jeffrey Brantingham**[2]      **Matthew Valasik**[3]      **Yevgeniy Vorobeychik**[1]

[1] Washington University in St. Louis
[2] University of California, Los Angeles
[3] Louisiana State University
{sixie.yu, yvorobeychik}@wustl.edu, branting@ucla.edu, mvalasik@lsu.edu

## Abstract

Network games are a natural modeling framework for strategic interactions of agents whose actions have local impact on others. Recently, a multi-scale network game model has been proposed to capture local effects at multiple network scales, such as among both individuals and groups. We propose a framework to learn the utility functions of binary multi-scale games from agents' behavioral data. Departing from much prior work in this area, we model agent behavior as following logit-response dynamics, rather than acting according to a Nash equilibrium. This defines a generative time-series model of joint behavior of both agents and groups, which enables us to naturally cast the learning problem as maximum likelihood estimation (MLE). We show that in the important special case of multi-scale linear-quadratic games, this MLE problem is convex. Extensive experiments using both synthetic and real data demonstrate that our proposed modeling and learning approach is effective in both game parameter estimation as well as prediction of future behavior, even when we learn the game from only a single behavior time series. Furthermore, we show how to use our framework to develop a statistical test for the existence of multi-scale structure in the game, and use it to demonstrate that real time-series data indeed exhibits such structure.

## 2 INTRODUCTION

A broad class of scenarios involving strategic interaction among a large collection of agents can be modeled by network (graphical) games, including investment in a public good [Bramoullé and Kranton, 2007; Grossklags et al., 2008], information diffusion [Galeotti et al., 2010], peer effects in social networks [Ballester et al., 2006], and adop-

tion of innovation [Jackson, 2010]. A prominent feature of network games is local effects, where an agent's utility depends only on the actions of its network neighbors [Kearns et al., 2001]. Many real networks, however, additionally exhibit group or community structure [Girvan and Newman, 2002], and Jin et al. [2021] recently proposed a multi-scale network game model that embeds such structure into the network game representation. However, a multi-scale game representation is often not given a priori, and instead what is available is time-series data of actual behavior, such as trade interactions among nations, or homicides arising from organized crime activities. Our goal is to develop a scalable framework for learning parametric models of multi-scale network games from such time-series data.

The general problem of learning utility functions in games from observed behavior has been extensively studied [Chajewska et al., 2001; Vorobeychik et al., 2007; Waugh et al., 2011; Honorio and Ortiz, 2015; Garg and Jaakkola, 2016; Leng et al., 2020]. A common assumption in this line of work is that agents are *fully rational* in that they act according to a Nash equilibrium. However, much experimental evidence suggests that this assumption is commonly violated [Andreoni and Miller, 1993; Camerer, 2003]. In addition, time-series behavior data often exhibits intertemporal dependence, such as the self-exciting nature of crime data [Mohler et al., 2011], a feature that is lost if behavior is modeled by a Nash equilibrium of a single-shot game.

We propose to use *logit-response dynamics (LRD)*—a classic framework to capture boundedly rational behavior in games [Blume, 1993; Alós-Ferrer and Netzer, 2010]—as a solution concept in learning utility functions from time-series data representing behavior in repeated strategic interactions. In LRD, each action by a player is played with a probability proportional to its utility, with actions of the other players fixed to what was played in the previous time step. LRD has two advantages over Nash equilibrium. First, it explicitly captures intertemporal dependence in behavior, since agents are responding to previously observed choices by others; in contrast, Nash equilibrium behavior in a one-

*Accepted for the 38th Conference on Uncertainty in Artificial Intelligence* (UAI 2022).

shot game exhibits no temporal dependence. Second, LRD solution concept is more psychologically plausible than Nash equilibrium behavior [Haile et al., 2008; Fudenberg et al., 1998; Stahl II and Wilson, 1994]. While Duong et al. [2010] also explicitly modeled intertemporal dependence in behavior, their approach was limited to consensus games, and required knowledge of utilities associated with player actions. Finally, ours is the first approach to consider multi-scale structure of strategic interactions on networks.

Armed with the game-theoretic generative model of time-series behavior data, we formulate the game learning problem as maximum likelihood estimation (MLE). In general, this problem can be (approximately) solved using gradient ascent; however, neither optimality nor consistency of estimation is guaranteed in our setting, where data is not generated i.i.d. To address this, we instantiate our framework in the context of parametric multi-scale linear-quadratic utility models. We prove that in this special case, the MLE problem is convex and can thus be solved efficiently. Our final technical contribution is a likelihood ratio test that enables us to statistically determine whether behavioral data generated by a multi-scale game model actually reflects multi-scale structure, where the null hypothesis is that only single-scale interactions significantly impact behavior.

We use extensive experiments on both synthetic and real datasets to demonstrate that the proposed approach effectively learns game parameters from time-series data. Furthermore, we show that our approach outperforms state-of-the-art baselines in predicting future agent behavior. Finally, we show that the game models we learn on real data offer interesting insights about behavior in the associated settings. For example, in the case of gang violence data, we show that the model we learn exhibits temporal self-excitation of homicides at multiple scales (that is, stemming from both individual gang member interaction, as well as interactions among gangs), generalizing insights from prior literature [Mohler et al., 2011]. The code to replicate the experiments is available at: https://github.com/marsplus/bMSGN.

In summary, our contributions are:

1. A novel framework for learning strategic agents' utility functions from behavioral data by modeling agent behavior using logit-response dynamics.
2. Support for learning multi-scale structure in agent utilities (i.e., strategic dependences among *groups* of agents). In addition to learning the utility functions, we propose a statistical test for the significance of multi-scale structure in utilities.
3. Experimental evaluation using real datasets demonstrating that the proposed approach outperforms prior art in predictive efficacy, and obtains useful insights about the associated domains.

**Related Work** Preference (or utility) elicitation, or inferring preferences of agents through active interaction, is a

classic problem in decision theory [Fischhoff and Manski, 2000; Blum et al., 2004]. The passive counterpart of preference elicitation is preference or utility learning from observed time-series data of behavior [Chajewska et al., 2001; Nielsen and Jensen, 2004]. Of direct relevance to our work is the literature on learning utility functions of players in game-theoretic models of their behavior. In this there are two major strands: learning utilities from observations of behavior time-series [Honorio and Ortiz, 2015; Garg and Jaakkola, 2016; Leng et al., 2020; Ling et al., 2018; Waugh et al., 2011], and learning utilities from observed payoffs [Duong et al., 2009; Vorobeychik et al., 2007; Gao and Pfeffer, 2010]. The principal difference between our framework and the former set of approaches stems from our use of LRD model of behavior, which considerably simplifies the learning problem and naturally allows us to capture temporal interdependence. Gao and Pfeffer [2010] use a closely related Quantal Response (QR) model of bounded rational behavior to learn game representations from data. However, this approach ignored temporal dependence, which is central to our framework. In addition, their approach assumed access to payoffs associated with player actions, whereas we make no such assumption. Our approach draws some inspiration from the framework for learning from collective behavior by Kearns and Wortman [2008]. However, the key general result in Kearns and Wortman [2008] requires learning with reset (i.e., a large collection of independently generated sequences of behavior), whereas we learn from only a single observed behavior sequence. Duong et al. [2010], like us, explicitly modeled intertemporal dependence in behavior. However, their approach was limited to consensus games, and required knowledge of player utilities.

## 3 MODEL

### 3.1 BINARY MULTI-SCALE GAME ON NETWORKS

A binary multi-scale game is defined on a network, which we represent by the adjacency matrix $\boldsymbol{A}$. The network can be directed or undirected, weighted or unweighted. We only assume that there are no self-loops in the network. For expository purposes, $\boldsymbol{A}$ is unweighted and undirected in the present paper. The agents in the game are situated on the vertices of $\boldsymbol{A}$, denoted by $\mathcal{V} = \{v_1, \ldots, v_n\}$, and are partitioned into $K$ groups, i.e., $\mathcal{V} = \bigcup_{j=1}^{K} \mathcal{G}_j$, and $\mathcal{G}_i \cap \mathcal{G}_j = \emptyset$ for any $i \neq j$. We use the set $\mathcal{J} = \{\mathcal{G}_j \mid j = 1, \ldots, K\}$ to represent the $K$ groups. Intuitively, we can use each group $\mathcal{G}_j$ to represent a neighborhood when the underlying network is an urban network, or an interest group if the underlying network is a social network. The group membership of agent $i$ is encoded by a mapping $\alpha(i)$ from the agent's index to its group index, i.e., $\alpha(i) = j$ for $i \in \mathcal{G}_j$. Throughout, we assume that the network structure $\boldsymbol{A}$, the mapping $\alpha(i)$, and the group structure $\mathcal{J}$ are known.

We use $x_i \in \mathcal{S}_i$ to represent agent $i$'s action, where $\mathcal{S}_i = \{0, 1\}$. We use public goods investment as a running example, where $x_i = 1$ (resp. $x_i = 0$) means that agent $i$ invests (resp. does not invest) in the public good. Consequently, we will refer to the choice $x_i = 1$ as an agent's decision to invest, while $x_i = 0$ means that $i$ decides not to invest. The marginal cost of making an investment is captured by a constant $c_i \in \mathbb{R}_+$, e.g., monetary cost, time, and/or effort exerted. The action profile of all agents is represented by $\boldsymbol{x} \in \{0, 1\}^n$, where the $i$-th entry is $x_i$. We use the set $\mathcal{N}(i)$ to represent agent $i$'s neighbors. The action profile restricted to agent $i$'s neighbors is $\boldsymbol{x}_{\mathcal{N}(i)}$.

To capture multi-scale (group) structure of the game, we define a vector $\boldsymbol{y} \in \mathbb{R}^K$, which represents some aggregate statistic at the group level. Typically, $y_j$ will be the total investment by group $j$, i.e., $y_j = \sum_{i \in \mathcal{G}_j} x_i$. We emphasize, however, that the definition of $\boldsymbol{y}$ is quite general, e.g., $y_j$ can also be the median investment from group $j$, or any other reasonable group-level statistic. The key idea behind the multi-scale representation is that while agents have concrete knowledge about the behavior of those they regularly interact with (network neighbors), they only have higher-level knowledge about other groups, as captured by the associated statistics for those groups. A concrete example is vaccination: an agent usually has more specific knowledge about the vaccination status of her close friends, which is encoded by $\boldsymbol{x}_{\mathcal{N}(i)}$, but only aggregate vaccination information at the level of counties or states, which is captured by $\boldsymbol{y}$. The utility function of agent $i$ is defined as follows:

$$u_i(x_i, \boldsymbol{x}_{-i}) = g_i\left(x_i, \boldsymbol{x}_{\mathcal{N}(i)}\right) + h_i(x_i, \boldsymbol{y}) - c_i x_i, \quad (1)$$

where $\boldsymbol{y}$ is implicitly a function of the full action profile $\boldsymbol{x}$. The function $g_i$ models local effects between an agent and its direct neighbors, capturing the externality that agent $i$ experiences from its neighbors' (and its own) investment. The function $h_i$ generalizes local effects from the individual level to the group level, encoding the multi-scale structure in the game. The term $c_i x_i$ captures the cost of investment. Putting everything together, we define a *binary multi-scale game on networks* as a tuple b-$\mathrm{MSGN}(\boldsymbol{A}, \mathcal{J}, \{\mathcal{S}_i\}, \{u_i\}_{i=1}^n)$, where $\boldsymbol{A}$ is the underlying network, $\mathcal{J}$ is the group structure, $\mathcal{S}_i$ are pure strategy sets of players, and $u_i$ are player utilities defined in Equation (1).

## 3.2 LOGIT-RESPONSE DYNAMICS

When modeling agents' strategic behavior, a common assumption is that agents are *rational*, i.e., they always choose the action with the highest utility. This is formally modeled by the best-response rule: $x_i \in \arg\max_{x_i'} u_i(x_i', \boldsymbol{x}_{-i})$. In the conventional Nash equilibrium solution concept that has been common in prior literature on learning games from data [Honorio and Ortiz, 2015; Leng et al., 2020], all players are assumed to simultaneously choose a best response

to each other. In reality, however, an agent may not make completely rational decisions, due to 1) limited resources or computational power needed to precisely solve the argmax problem and 2) inability to perfectly assess small differences in its utility. Furthermore, a Nash equilibrium of a static game cannot capture intertemporal dependencies that may be present in time-series behavior data, and multiplicity of equilibria creates a further practical challenge in learning general-sum games from data. A common alternative to the Nash equilibrium solution concept is a *quantal response equilibrium (QRE)* [McKelvey and Palfrey, 1995], which was recently used in a framework for learning *two-player zero-sum* games from data [Ling et al., 2018]. However, multiplicity of equilibria (both Nash and QRE) in general-sum games has limited further progress.

*Our key conceptual contribution is to combine bounded rationality in action choices with bounded rationality in dynamic agent behavior.* While such a combination seems entirely natural, we are the first to explore it in the context of learning games from time-series data. Our experiments below vindicate this approach, which resolves both the issue of multiplicity of equilibria and dynamic interdependencies in behavior. Specifically, we adopt a classic model of boundedly-rational dynamic behavior: *logit-response dynamics (LRD)* [Blume, 1993; Alós-Ferrer and Netzer, 2010]. LRD presumes a repeated one-shot game in which agents select actions with probabilities proportional to their utilities (as in QRE) in every step, taking choices made by others as given from the previous step (*unlike* QRE). In our context, the probability of agent $i$ choosing to invest ($x_i = 1$) in the next time step is

$$
\begin{aligned}
p(x_i^{t+1} = 1 \mid \boldsymbol{x}^t, \boldsymbol{y}^t) &= \frac{e^{\gamma \cdot u_i(1, \boldsymbol{x}_{-i}^t, \boldsymbol{y}^t)}}{e^{\gamma \cdot u_i(1, \boldsymbol{x}_{-i}^t, \boldsymbol{y}^t)} + e^{\gamma \cdot u_i(0, \boldsymbol{x}_{-i}^t, \boldsymbol{y}^t)}} \\
&= \frac{1}{1 + e^{\gamma\left(u_i(0, \boldsymbol{x}_{-i}^t, \boldsymbol{y}^t) - u_i(1, \boldsymbol{x}_{-i}^t, \boldsymbol{y}^t)\right)}}.
\end{aligned}
\tag{2}
$$

The scalar $\gamma$ quantifies the noise level in the agent's decision-making. As $\gamma$ goes to infinity, the logit-response converges to the best-response rule. For any $0 < \gamma < \infty$, the agent chooses a non-best response with positive probability, and the actions yielding larger utility are chosen with higher probability. Throughout the paper, we assume that $\gamma$ is known. We define the probability $p(x_i^{t+1} = 1 \mid \boldsymbol{x}^t, \boldsymbol{y}^t)$ as the *investment probability* at time step $t + 1$. When the context is clear we use $p(x_i^{t+1})$ to represent the investment probability, omitting the dependence on $\boldsymbol{x}^t$ and $\boldsymbol{y}^t$.

In LRD, we assume that at each time step each agent updates its action independently according to the logit response function (2). Consequently, given $\boldsymbol{x}^t$ and $\boldsymbol{y}^t$ the agents' investment decision at time step $t + 1$ are conditionally independent, i.e., $x_i^{t+1}$ and $x_j^{t+1}$ are independent for $i \neq j$. Additionally, this assumption implies convergence of agents' behavior to a stationary distribution. Specifically, let $\mathcal{M}$ be

the discrete Markov chain induced from the logit-response dynamics, with state space $\mathcal{S} = \{0,1\}^n$. The transition probability $p(\boldsymbol{x}^{t+1}|\boldsymbol{x}^t)$ equals $\prod_{i=1}^{n} p(x_i^{t+1} = 1 \mid \boldsymbol{x}^t, \boldsymbol{y}^t)$, which by definition is always positive, including the transition probability from a state to itself. Consequently, the state transition graph of $\mathcal{M}$ is strongly connected and aperiodic.[1] This in turn implies that the stationary distribution $\pi$ of the Markov chain exists and is unique [Chung and Graham, 1997; Wildstrom, 2005].

## 4 THE LEARNING FRAMEWORK

Since in practice we typically only have a single trail of past behavior to learn from, we consider the problem of learning a game model parameters from a single behavior sequence collected over $l$ time steps, i.e., $\mathcal{D}_l = \{(\boldsymbol{x}^1, \boldsymbol{y}^1), \ldots, (\boldsymbol{x}^l, \boldsymbol{y}^l)\}$, where $\boldsymbol{x}^t$ is the action profile of all agents at time step $t$ and $\boldsymbol{y}^t$ is the group-level statistics that capture aggregate behavior by each group in the multi-scale game. We assume that the utility functions of players $u_i$ have parametric representations, with associated parameter vectors denoted by $\boldsymbol{\theta}_i \in \mathcal{F}_i := [-1, 1]^m$, where $m$ is the dimension of $\boldsymbol{\theta}_i$; these are concatenations of the parameters of $g_i$ and $h_i$ (and the cost $c_i$), the two main constituent functions in player utilities. We use $\Theta = \{\boldsymbol{\theta}_1, \ldots, \boldsymbol{\theta}_n\}$ to represent all learnable parameters of the game, where $\Theta \in \Pi = \mathcal{F}_1 \times, \ldots, \times \mathcal{F}_n$. The utility function in (1) is a high-level description; we will instantiate $g_i$ and $h_i$ to specific parametric functions below. We present a general likelihood-based approach for learning multi-scale games from such data, and subsequently study an important special case which admits efficient learning.

### 4.1 THE GENERAL CASE

The binary multi-scale game together with the logit-response dynamics define a generative time-series model of joint behavior of both agents and groups. We assume that $\boldsymbol{y}^t$ is a deterministic function of the individual-level action profile $\boldsymbol{x}^t$, which simplifies the derivation of the data likelihood, as the joint probability of $\boldsymbol{x}^{t+1}$ and $\boldsymbol{y}^{t+1}$ reduces to the marginal probability of $\boldsymbol{x}^{t+1}$. The generative model is a discrete Markov chain over action profiles. Omitting the dependence of the investment probability on $\boldsymbol{x}^t$ and $\boldsymbol{y}^t$, the data likelihood $\mathcal{L}(\mathcal{D}_l; \Theta)$ is formulated as follows:

$$\mathcal{L}(\mathcal{D}_l; \Theta) = p(\boldsymbol{x}^1) \prod_{t=1}^{l-1} p(\boldsymbol{x}^{t+1}|\boldsymbol{x}^t, \boldsymbol{y}^t) =$$

$$\prod_{t=1}^{l-1} \prod_{i=1}^{n} \left[ p(x_i^{t+1} = 1) \right]^{x_i^{t+1}} \left[ 1 - p(x_i^{t+1} = 1) \right]^{1 - x_i^{t+1}},$$

$$(3)$$

---

[1]The state transition graph of a discrete Markov chain is aperiodic if the transition probability from a state to itself is positive.

where the last equality utilizes the assumption that $x_i^{t+1}$ and $x_j^{t+1}$ are independent given $\boldsymbol{x}^t$ and $\boldsymbol{y}^t$, and the fact that $p(\boldsymbol{x}^1) = 1$. We learn the parameters $\Theta$ by resorting to the maximum likelihood estimation (MLE). In general, we can leverage gradient-based methods and automatic differentiation tools to maximize the likelihood, as long as the utility functions are differentiable.

With a slight abuse of notation, we use b-MSGN($\Theta$) to represent the generative model (consisting of the game together with the logit-response dynamics solution concept), with the utility functions parameterized by $\Theta$. We now instantiate the utility function to a specific parametric form. In particular, we consider games with *linear-quadratic utility functions*, augmented with the $h_i$ to account for the multi-scale structure. The resulting MLE problem is convex, and can thus be (near-)optimally solved using interior point methods. We also develop a statistical test for the existence of multi-scale structure in this game based on the classic likelihood ratio test.

### 4.2 LEARNING MULTI-SCALE LINEAR-QUADRATIC GAMES

*Linear-quadratic games* have been used in much prior literature on network game modeling both in economics and machine learning [Ballester et al., 2006; Bramoullé and Kranton, 2007; Galeotti et al., 2020; Leng et al., 2020], with Leng et al. [2020] specifically considering the problem of learning network structure in such models from Nash equilibrium behavior by the agents. The standard utility function in linear-quadratic network games is defined as

$$u_i(x_i, \boldsymbol{x}_{-i}) = b_i x_i + \beta_i x_i \sum_{j \in \mathcal{V}} A_{i,j} x_j - c_i x_i^2, \quad (4)$$

where $b_i \geq 0$ is the marginal benefit of investing, $c_i \geq 0$ is the cost to invest, and $\beta_i \in \mathbb{R}$ captures peer effects from the neighbors' investment. When $\beta_i > 0$ (resp. $\beta_i < 0$), higher investment from the neighbors encourages agent $i$ to make more (resp., less) investment.

To model the multi-scale structure in the game, we consider the following group-level aggregate function $h_i$:

$$h_i(x_i, \boldsymbol{y}) = \eta_i x_i \left( y_{\alpha(i)} - \frac{\sum_{g \in \mathcal{J} \setminus \{\mathcal{G}_{\alpha(i)}\}} y_g}{|\mathcal{J}| - 1} \right), \quad (5)$$

where the second term in the parentheses is the average of the statistics from other groups. The difference models the relative magnitude of the statistics between agent $i$'s group and other groups. When $\eta_i > 0$ (resp., $\eta_i < 0$), higher relative investment by agent $i$'s group compared to other groups encourages (resp., discourages) $i$'s own investment.

We augment the linear-quadratic payoff with the function

$h_i$, leading to the *multi-scale linear-quadratic utility*:

$$u_i(x_i, \boldsymbol{x}_{-i}) = (b_i - c_i)x_i + \beta_i x_i \sum_{j \in \mathcal{V}} A_{i,j}x_j + h_i(x_i, \boldsymbol{y}).$$

(6)

The set $\boldsymbol{\theta}_i = \{b_i, \beta_i, \eta_i, c_i\}$ consists of the parameters we aim to learn from data. Note that as the action space in our setting is binary, the term $b_i x_i - c_i(x_i)^2$ becomes $(b_i - c_i)x_i$. As a result, accurately estimating the two parameters may not be feasible, as they can be shifted the same amount without changing the difference.[2] Therefore, we treat $b_i - c_i$ as a single *marginal benefit* that we estimate from data.

As we now show, the key property of this multi-scale linear quadratic game model is that the resulting MLE problem is convex. The proof is a standard argument of showing convexity by leveraging second order derivatives.

**Proposition 4.1.** *Consider a* b-MSGN$(\boldsymbol{A}, \mathcal{J}, \{u_i\}_{i=1}^n)$. *If* $\{u_i\}_{i=1}^n$ *are instantiated as the multi-scale linear-quadratic utilities, the resulting MLE optimization problem is convex.*

*Proof.* Recall that $\Theta \in \Pi = \mathcal{F}_1 \times, \ldots, \times \mathcal{F}_n$, that is, a Cartesian product of a set of convex sets. Thus, the feasible region $\Pi$ of the MLE is convex. In what follows, we show that the log-likelihoof function $\log \mathcal{L}(\mathcal{D}_l; \Theta)$ is concave w.r.t. $\Theta$.

Note that $\log \mathcal{L}(\mathcal{D}_l; \Theta) = \sum_{t=1}^{l-1} \log p(\boldsymbol{x}^{t+1}|\boldsymbol{x}^t)$; it is sufficient to show that $\log p(\boldsymbol{x}^{t+1}|\boldsymbol{x}^t)$ is concave w.r.t. $\Theta$ for any $1 \le t \le l-1$. We expand $\log p(\boldsymbol{x}^{t+1}|\boldsymbol{x}^t)$ as follows:

$$\log p(\boldsymbol{x}^{t+1}|\boldsymbol{x}^t) = \sum_{i=1}^n \left[ x_i^{t+1} \log p(x_i^{t+1} = 1) + \right.$$
$$\left. (1 - x_i^{t+1}) \log[1 - p(x_i^{t+1} = 1)] \right],$$

The logarithm of the investment probability is as follows:

$$\log p(x_i^{t+1} = 1) = \log \left[ \frac{1}{1 + e^{-\gamma \cdot u_i(1|\boldsymbol{x}^t, \boldsymbol{y}^t, \boldsymbol{\theta}_i)}} \right].$$

It is direct that $u_i(1|\boldsymbol{x}^t, \boldsymbol{y}^t, \boldsymbol{\theta}_i)$ is a linear function of $\boldsymbol{\theta}_i$. In addition, $\log p(x_i^{t+1} = 1)$ is concave w.r.t. $u_i(1|\boldsymbol{x}^t, \boldsymbol{y}^t, \boldsymbol{\theta}_i)$, as the second derivative is negative over the domain, i.e.,

$$\frac{\partial^2 \log p(x_i^{t+1} = 1)}{\partial^2 u_i(1|\boldsymbol{x}^t, \boldsymbol{y}^t, \boldsymbol{\theta}_i)} = -\frac{e^{\gamma \cdot u_i(1|\boldsymbol{x}^t, \boldsymbol{y}^t, \boldsymbol{\theta}_i)} \cdot \gamma^2}{(1 + e^{\gamma \cdot u_i(1|\boldsymbol{x}^t, \boldsymbol{y}^t, \boldsymbol{\theta}_i)})^2} < 0.$$

The composition of a linear function with a concave function leads to a concave function (Chapter 3.2.2 of [Boyd and Vandenberghe, 2004]); thus, $\log p(x_i^{t+1} = 1)$ is concave w.r.t.

---

[2]This problem is not specific to our model: in prior literature, the cost constant $c_i$ is usually set to $\frac{1}{2}$ in order to avoid the invariance of $b_i - c_i$ to the shifting.

$\boldsymbol{\theta}_i$. We can similarly show that $\log[1 - p(x_i^{t+1} = 1)]$ is convex w.r.t. $\boldsymbol{\theta}_i$, which implies that $(1 - x_i^{t+1}) \log p(x_i^{t+1} = 1)$ is concave w.r.t. $\boldsymbol{\theta}_i$. A linear combination of concave functions is concave, so $\log p(\boldsymbol{x}^{t+1}|\boldsymbol{x}^t)$ is concave w.r.t. $\Theta$. $\square$

**A Statistical Test for Multi-Scale Structure** We now further leverage the proposed framework to develop a statistical test to check whether the game exhibits multi-scale structure. This test is based on the classic *likelihood ratio test* [Wasserman, 2013]. Specifically, let $\hat{\Theta} = \{\hat{\boldsymbol{b}}, \hat{\boldsymbol{c}}, \hat{\boldsymbol{\beta}}, \hat{\boldsymbol{\eta}}\}$ be the MLE estimator. The feasible region of $\hat{\Theta}$ is $\mathcal{F} = \{\hat{\Theta} \mid \hat{\boldsymbol{b}} \ge 0, \hat{\boldsymbol{c}} \ge 0, \hat{\boldsymbol{\beta}} \in [-\boldsymbol{1}, \boldsymbol{1}], \hat{\boldsymbol{\eta}} \in [-\boldsymbol{1}, \boldsymbol{1}]\}$. The null hypothesis set is $\mathcal{F}_0 = \{\hat{\Theta} \in \mathcal{F} \mid \hat{\boldsymbol{\eta}} = \boldsymbol{0}\}$, encoding the hypothesis that group-level statistics have no impact on agents' utilities. The test statistic is as follows:

$$\lambda = 2 \log \left( \frac{\max_{\Theta \in \mathcal{F}} \mathcal{L}(\mathcal{D}_l; \Theta)}{\max_{\Theta \in \mathcal{F}_0} \mathcal{L}(\mathcal{D}_l; \Theta)} \right).$$

(7)

Intuitively, $\lambda$ is large if there is some estimator $\hat{\Theta}$ in the feasible region $\mathcal{F}$ for which the data $\mathcal{D}_l$ is much more likely than for any estimator in the null hypothesis set $\mathcal{F}_0$. The p-value equals $p(\chi_n^2 > \lambda)$, where $\chi_n^2$ follows a chi-square distribution with $n$ degrees of freedom [Wasserman, 2013]. In the Experiments section, we present experiments on synthetic data to show that the test is indeed effective at identifying multi-scale structure in games. We then use it on real data to demonstrate that such data also exhibits statistically significant multi-scale behavior dependence.

## 5 EXPERIMENTS

We focus our experimental study on learning a multi-scale linear-quadratic game b-MSGN$(\Theta^*)$. In all cases, we learn the game from a sequence $\mathcal{D}_l$, and experiment on both synthetic and real-world data. We use synthetic data to demonstrate the effectiveness of our approach at *recovering the groundtruth parameters* of the linear-quadratic games, and additionally show that the statistical test successfully identifies multi-scale game structure.

In addition, we evaluate the efficacy of the proposed approach to predict future time-series behavior. For both synthetic and real data, we first compare predictive efficacy of the proposed game learning approach with three conventional generative baseline approaches commonly applied in similar settings with the primary purpose of time-series prediction: a discrete Markov chain, a homogeneous Poisson process, and the Hawkes process [Mohler et al., 2011]. Specifically, our experiments use a discrete-time Hawkes process with exponential decay function; the intensity function at time step $t$ is: $\lambda(t) = \lambda_0 + \alpha \sum_{t_i < t} z_{t_i} e^{-\beta(t-t_i)}$; $\lambda_0$ and $\alpha$ are estimated through MLE; $\beta$ is selected by cross-validation; $z_{t_i}$ is the sum of $\boldsymbol{x}^{t_i}$, i.e., $\sum_{j=1}^n x_j^{t_i}$. We show that the proposed approach outperforms these baselines in terms of prediction accuracy.

Additionally, we compare our approach with a method for learning *Linear Influence Games* (LIGs) [Honorio and Ortiz, 2015], a state-of-the-art game-theoretic baseline for learning utility functions from time-series behavior in network games. LIG is a generative model that assumes that behavior in each step in a time-series is generated according to a mixture of two distributions: a uniform distribution over the set of all pure-strategy Nash equilibria, and a uniform distribution over the set of all non-equilibrium strategy profiles.[3] The learnable parameters of an LIG include the parameters of the players' utility functions as well as a parameter deciding which distribution an action profile comes from. The parameters are learned by maximizing the proportion of equilibria observed in the training data.

## 5.1 SYNTHETIC DATA

We generate a synthetic sequence $\mathcal{D}_l$ by simulating b-MSGN($\Theta^*$) for $l-1$ iterations, with starting action profile initialized as zeros. In each time step, every agent makes a decision according to the Bernoulli distribution with success rate equal to the investment probability (i.e., Equation (2)). The ground-truth parameters $\Theta^* = \{\boldsymbol{b}^*, \boldsymbol{c}^*, \boldsymbol{\beta}^*, \boldsymbol{\eta}^*\}$ are specified as follows: $b_i^* \sim \mathcal{N}(0.3, 0.01^2)$, $c_i^* \sim \mathcal{N}(1.3, 0.1^2)$, $\beta_i^* \sim \mathcal{N}(-1, 0.01^2)$ and $\eta_i^* \sim \mathcal{N}(0.1, 0.01^2)$. The parameter $\gamma$ is set to 5. We consider three classes of synthetic networks: Barabási-Albert (BA) [Barabási and Albert, 1999], Watts-Strogatz (WS) [Watts and Strogatz, 1998], and Block Two-level Erdős-Rényi (BTER) [Seshadhri et al., 2012] networks. For each class, we randomly generate 20 networks with 100 nodes each. For each randomly generated network, we run the community detection algorithm proposed by Clauset et al. [2004] and use the resulting communities as groups.

Figure 1 shows the effectiveness of learning the game parameters from synthetic data. As the length $l$ increases, the Root Mean Squared Error (RMSE) between the estimated parameters and the true parameters consistently decreases, converging to near-zero; this indicates that the MLE estimator approximates the ground-truth $\Theta^*$ reasonably well.

Next, we show that the statistical test successfully determines the existence of the multi-scale structure in the game. We simulate two sets of data, one is called "with groups" and the other "without group". The "with groups" data is simulated as usual, such that the agents' utilities are influenced by the multi-scale structure. The "without group" data is simulated with $\eta_i^*$ set to zero, which implies that the multi-scale structure does not have a direct impact on the agents' utilities. The p-values for the two sets of data are shown in Figure 2. The red horizontal lines represent where

---
[3]Note that the LIG approach assumes that the set of all pure-strategy Nash equilibria can be efficiently sampled. Another advantage of the proposed approach over LIG is that we do not need this assumption.

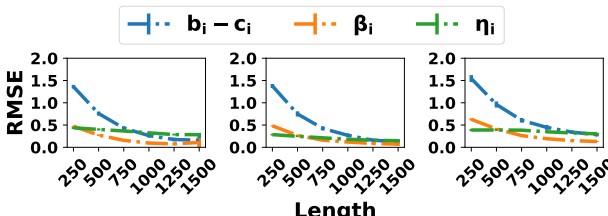

Figure 1: The RMSE between the estimated parameters and the true parameters across various lengths $l$. **Left**: BA (averaged degree=5.82, averaged clustering coeff.=0.1067); **Middle**: WS (averaged degree=9.1064, averaged clustering coeff.=0.3542); **Right**: BTER (averaged degree=9.3200, averaged clustering coeff.=0.1299).

$p(\chi_n^2 > \lambda) = 0.05$: we reject the null hypothesis when the p-value is below the red line. The blue lines represent the p-values for the "with groups" data. We can see that as $l$ (the number of observations) increases the p-values consistently decrease. In particular, for BA and SW networks when $l > 750$ we correctly reject the null hypothesis. The dashed orange lines represent the p-values for the "without group" data. Note that the orange lines are consistently above 0.05 by a large margin, which means that we never incorrectly reject the null hypothesis (i.e., never falsely claim the existence of multi-scale structure).

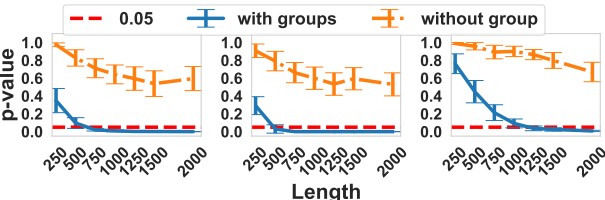

Figure 2: Experimental results for the statistical test. The blue solid lines (resp. orange dashed lines) represent the p-values evaluated on the data with (resp. without) the multi-scale structure. **Left**: BA; **Middle**: WS; **Right**: BTER.

## 5.2 REAL-WORLD DATA

**Gang-Related Homicides.** We learn the game on gang-related homicides data from Los Angeles [Valasik et al., 2017]. The data includes 1425 incidents from 1978 to 2012. Each incident consists of several attributes, including date, address, coordinates ($X$ and $Y$ correspond to latitude and longitude, respectively), and demographic information of the victim and the suspect. Each incident includes a label indicating whether the homicide is gang-related, and if so, includes an attribute of the suspect's gang affiliation. All sensitive attributes in the experimental results are anonymized with numerical values. The data is preprocessed as follows. First, we keep only the incidents that are gang-related, and discard the incidents with missing attributes. Second, to

correct errors in incident coordinates, we compute the geometric center of the incidents' coordinates, and then fit a standard Gaussian distribution on their distances to the center, and finally discard any incidents that are three standard deviation away from the center. After preprocessing, the data contains 606 incidents committed by suspects from 54 gangs. A gang's location is approximated by the geometric center of its associated incidents. We treat the 54 gangs as the agents in the game; they are partitioned into three groups according to their neighborhood information. The network $A$ is weighted, undirected, and complete, with the gangs as nodes. The weight on an edge is the inverse of the driving time between the two endpoints (gangs) obtained by querying the Google Maps API.

Next, we construct a sequence $\mathcal{D}_l$ of action profiles from the processed data by discretizing time and grouping incidents that occur in each time interval, where $T$ is the hyperparameter corresponding to the length of the interval in days (i.e., how finely the data is discretized). We experiment with different values of $T$, i.e., $T = 30, 60, 90, 120, 150, 180, 240, 365$. We set $x_j^t = 1$ if there is at least one incident associated with the $j$-th gang at time step $t$, and set $x_j^t = 0$ otherwise. The aggregate statistic $y_i^t = \sum_{j \in \mathcal{G}_i} x_j^t$, measures the overall level of violence in group $\mathcal{G}_i$.

We first apply the statistical test on data aggregated with different values of $T$. The p-values are less than 0.05 across the values of $T$, except for $T = 30$ and 120. The overall observation is that the data consistently exhibits statistically significant multi-scale behavior dependence, an effect that is relatively robust to time discretization; the only instances where its influence is not statistically significant is for $T = 30$ and 120.

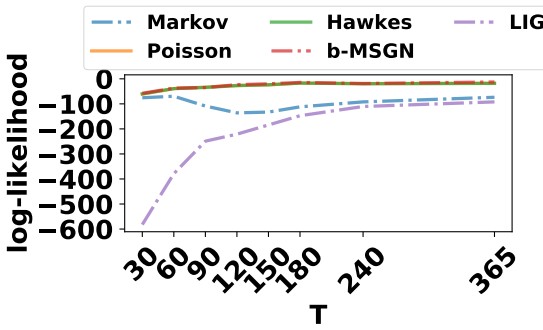

Figure 3: Comparison of our approach with the game-theoretic baseline LIG and three conventional generative approaches in terms of predictive log-likelihood on test data.

To compare the proposed approach, in which we learn the linear-quadratic game on this data, with several baselines in terms of predictive log-likelihood on test data, we split $\mathcal{D}_l$ into training data and test data with ratio $9 : 1$. The results are shown in Figure 3. We observe that our approach is

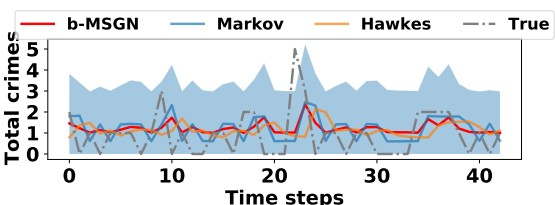

Figure 4: A visualization of the predicted total crimes on test data with $T = 30$ (i.e., each time step represents 30 days). We omit Poisson and LIG as their predictions are far from the ground-truth. The shaded area represents two standard deviations of the prediction from b-MSGN.

considerably better than LIG, particularly for smaller values of $T$. In addition, our approach is competitive in predictive accuracy with all but the Markov chain baseline (which is considerably worse), including the Hawkes process, which is the state-of-the-art approach for modeling crime data of this kind [Mohler et al., 2011].

A visualization of the predicted total crimes on test data is shown in Figure 4; the shaded area represents two standard deviations of the prediction from b-MSGN. The predictions from Poisson and LIG are omitted as they are far from the groundtruth; both are almost horizontal lines without capturing any trends exhibited in real data. We can observe that b-MSGN is capturing the overall trend with high confidence, i.e., the ground-truth lies within two standard deviations of the prediction.

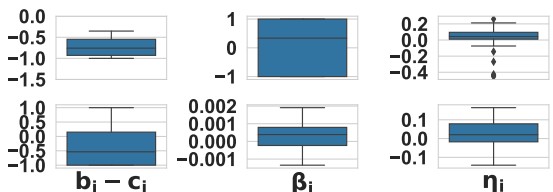

Figure 5: The estimates of $b_i - c_i$, $\beta_i$ and $\eta_i$. **Top**: the homicides data aggregated with $T = 60$. **Bottom**: the bilateral trading data.

The key advantage of the proposed approach comes from its interpretability as capturing strategic interactions, and in linear-quadratic games in particular, the parameters we learn have a natural interpretation, which we now consider. Specifically, to analyze the game parameters we have learned, we set $T = 60$ as an illustration (the results are quite robust to this), so that the resulting sequence $\mathcal{D}_l$ has $l = 213$ time steps. As we do not have access to the ground-truth utility functions, the analysis serves to provide insights about the gangs' behavior. The learned parameters are shown in the top row of Figure 5. First, the estimated $b_i - c_i$ are shown on the left of the figure; the median is $-0.77$. Note that the estimates are negative, that is, perceived costs of homicides by gang members exceed benefits. Overall, gang-related

homicides are relatively rare; indeed on average, only $4.7\%$ gangs that committed homicides in each time step; when increasing $T$ to 365, there are on average $19.7\%$ gangs that committed homicides in each time step and the median of $b_i - c_i$ becomes $-0.56$.

The estimates of $\beta_i$ are shown in the middle of the figure. The mean is $0.18$, which indicates that gang members on average tend to commit more homicides as the number of homicides from other members of their gang increases. This may be explained by the self-excitation phenomenon observed by Mohler et al. [2011] that an incident involving rival gangs can lead to retaliatory acts of homicide. Finally, the estimated $\eta_i$ are shown on the right of the figure. Most estimates are positive (except a few outliers), which suggests an intuitive observation that a greater overall level of violence in a gang's neighborhood tends to lead to greater incidence of violence by the gang.

To see how the discretization affects the estimates, we plot the estimated parameters across the values of $T$ as in Figure 6. The estimates of $b_i - c_i$ increase as $T$ gets larger. The estimates of $\beta_i$ and $\eta_i$ are also affected by the values of $T$. This suggests that the interpretation of the estimate has to consider the specific value of $T$. Indeed, the discretization changes the generative process of the data that is used to train the model. A future research question is to decide the optimal discretization in terms of a quantitative measure.

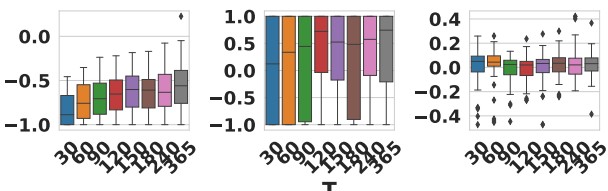

Figure 6: From left to right, the estimations of $b_i - c_i$, $\beta_i$ and $\eta_i$ across different values of $T$; the feasible region of each estimated parameter is restricted to $[-1, 1]$.

**Bilateral Trading Data.** The second dataset we consider is the bilateral trading data from the United Nations Comtrade Database (https://comtrade.un.org/). The data consists of statistics for international bilateral trading (e.g., imports and exports), including over 170 reporting economies and records from 1962 to 2018. We focus on annual exports data in terms of their value in US-dollars and extract a subset consisting of 127 reporting economies with complete statistics since 1962; the reporting economies are partitioned into six groups according to the continents they are located on: Asia, Africa, Europe, South America, Australia and North America. We treat the reporting economies as agents in the game. The graph underlying the game is directed and weighted, where an edge from $i$ to $j$ means that $i$ has exported goods/service to $j$, and the weight on the edge is the normalized total value of exports since 1962. As the graph is directed, we define the neighborhood of economy $i$ as its exporting destinations. The sequence $\mathcal{D}_l$ of action profiles consists of 57 time steps, each corresponding to a year. For every economy, we track a moving average of the value of exports over $k$ time steps. Let $e_i^t$ be the value of exports of economy $i$ at time step $t$. For $t > k$, if the value is greater than the moving average, i.e., $e_i^t > (e_i^{t-1}+, \ldots, +e_i^{t-k})/k$, we set $x_i^t = 1$; otherwise $x_i^t = 0$. For $t = 1, \ldots, k$ the actions $x_i^t$ are always set to zero. Intuitively, $x_i^t = 1$ encodes that economy $i$ has a higher value of exports compared with the average value of the previous three years, which signals economic growth [Michaely, 1977]. The group-level statistic is again $y_i^t = \sum_{j \in \mathcal{G}_i} x_j^t$. We experiment with five values of $k$, ranging from 1 to 5.

We first run the statistical test on $\mathcal{D}_l$. The resulting p-values are nearly zero across all the values of $k$, providing strong evidence to reject the null hypothesis (i.e., $\boldsymbol{\eta} = \mathbf{0}$). Therefore, a b-MSGN with $\boldsymbol{\eta} \neq \mathbf{0}$ better explains the data in terms of likelihood, which supports introducing the multi-scale structure into the game.

Next, we compare the game with the baselines on test data (the last $15\%$ of the entire sequence) in terms of predicted log-likelihoods. The results for $k = 5$ are as follows: 1) Markov Chain: $-55.2620$, 2) Poisson: $-74.1376$, 3) Hawkes: $-63.9631$, 4) LIG: $-51.2281$, and 5) b-MSGN: $-40.1436$; the results for other values of $k$ are similar.

Finally, the estimated parameters are shown in Figure 5 (second row). The estimated $b_i - c_i$ are mostly negative, indicating that for most economies it is difficult to maintain a steady growth in exports. Most estimated values of $\beta_i$ are positive, suggesting that an economy will have a growth in exports when its exporting destinations also have increasing exports. Finally, most estimated values of $\eta_i$ are positive, which suggests that the relative growth of a group's exports (compared with other groups) is a good predictor of the participating economies' growth.

To study the sensitivity of the estimated parameters to $k$, we plot the estimated parameters across the values of $k$ in Figure 7. The conclusion is similar to what we had for Figure 6: the estimated parameters are affected by $k$ and the interpretation of the estimate has to consider the specific value of $k$.

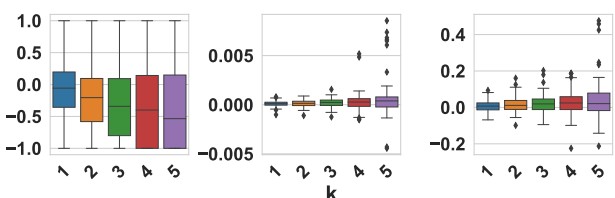

Figure 7: From left to right, the estimations of $b_i - c_i$, $\beta_i$ and $\eta_i$ across different values of $k$; the feasible region of each estimated parameter is restricted to $[-1, 1]$.

# 6 CONCLUSION

We propose a game-theoretic generative model of time-series behavior data by combining single-shot multi-scale network games with logit-response dynamics. We do not assume that the agents are fully rational, but rather that they make decisions according to logit-response dynamics. We then present a general learning framework based on maximum likelihood estimation (MLE) for inferring parameters of such games. In the special case of multi-scale linear-quadratic games we prove that the MLE is a convex optimization problem and thus admits efficient solution algorithms. We further develop a statistical test to determine whether the game exhibits multi-scale structure. We use extensive experiments on both synthetic and real datasets to show the efficacy of the proposed approach.

Our work considers aggregated statistics $y^t$ as deterministic w.r.t. the individual-level action profile $x^t$. However, it would be more realistic to model $y^t$ as a probabilistic function of $x^t$ due to the noise from the aggregation process. The probabilistic modeling complicates the derivation of the data likelihood since we need to have a joint distribution of $x^t$ and $y^t$. Another future direction is to consider more general multi-scale structures than the simple difference as studied in Section 4.2. Finally, the group structures $\mathcal{J}$ and the group memberships $\alpha(i)$ may not available in practice; one way to generalize the current model is to jointly learn $\mathcal{J}$ and $\alpha(i)$ from data.

**Acknowledgments** This research was supported in part by the National Science Foundation (grants IIS-1905558 and IIS-1903207), Army Research Office (MURI grant W911NF1810208), and NVIDIA.

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
