# OpenReview forum: "Learning Binary Multi-Scale Games on Networks"
_auai.org/UAI/2022/Conference — UAI 2022 Poster_

### Official Review · Reviewer_j2md · 2022-04-02

**Q2(1) Originality/Novelty:** 3
**Q2(2) Significance/Impact:** 3
**Q2(3) Correctness/Technical Quality:** 3
**Q2(6) Clarity Of Writing:** 3
**Q6 Overall Score:** 6
**Q8 Confidence In Your Score:** 4

**Q1 Summary And Contributions:**

This article presents an approach to learn the parameters of a multiplayer game of the family of binary multi-scale games. The authors assume that behavioural data is generated by players following a Logit-Response Dynamics.
The main interest of adopting such a model is that the optimization problem resulting from the obseerved strategies sequence likelihood maximization is convex.
Experimental results s on synthetic and real data are provided to illustrate the approach.

**Q10 Ethical Concerns (Optional):**

None.

**Q2 Assessment Of The Paper:**

More detailed information regarding each of these aspects is given below:

**Q2(4) Quality Of Experiments (Optional):**

3: Good: The experimental evaluation is adequate, and the results convincingly support the main claims.

**Q2(5) Reproducibility:**

2: Fair: Key resources (e.g., proofs, code, data) are unavailable but key details (e.g., proof sketches, experimental setup) are sufficiently well-described for an expert to confidently reproduce the main results.

**Q3 Main Strengths:**

The paper is clearly written the results well-presented and the contribution is substantial : Learning is modelled as a convex optimization problem.
The approach seems possible to reproduce on data (even though code is not available and experiments not reproducible).

**Q4 Main Weakness:**

Assumptions on game structure (local-global and linear quadratic utilities) are very specific. So is the assumption of LRD behaviour.
In total, the result is nice from a theoretical point of view, but not much is written on how to apply it in practice and to test the assumptions.
Finally, it is a serious weakness that code and experiments are not made available...

**Q5 Detailed Comments To The Authors:**

The paper is well-written and the result interesting, from a theoretical point of view.
Despite the very specific (and quite unusual) models of utilities and generative models, I find it interesting.
The only serious weakness of the paper is the non-availability of the code and the non-reproducibility of results.
All the more as this paper cannot be qualified as "theoretical": There is only one theoretical result, which is not especially difficult. So, it seems to me necessary to make the supproting codes available...


**Q7 Justification For Your Score:**

In favor of acceptation: The fact that the paper provides new results is easy to read and that the proposed learning model/algorithm may be useful.
The limits are its theoretical content which is limited/specific, which is not counterbalanced by making code available to the community.

**Q9 Complying With Reviewing Instructions:**

1: Yes.

---

### Official Review · Reviewer_1Xca · 2022-04-14

**Q2(1) Originality/Novelty:** 3
**Q2(2) Significance/Impact:** 3
**Q2(3) Correctness/Technical Quality:** 2
**Q2(6) Clarity Of Writing:** 2
**Q6 Overall Score:** 4
**Q8 Confidence In Your Score:** 3

**Q1 Summary And Contributions:**

This paper propose a framework to learn the utility functions of binary multi-scale games from agents' behavioral data. Distinguished from previous work in this area, the authors model the agent behavior as logit-response dynamics, rather than a Nash equilibrium. After that, the authors formulate the game learning problem as MLE and prove that in the special case, i.e. with parametric multi-scale linear-quadratic utility models, the MLE problem is convex and can be solved efficiently.

**Q2 Assessment Of The Paper:**

More detailed information regarding each of these aspects is given below:

**Q2(4) Quality Of Experiments (Optional):**

2: Fair: The experimental evaluation is weak: important baselines are missing, or the results do not adequately support the main claims.

**Q2(5) Reproducibility:**

2: Fair: Key resources (e.g., proofs, code, data) are unavailable but key details (e.g., proof sketches, experimental setup) are sufficiently well-described for an expert to confidently reproduce the main results.

**Q3 Main Strengths:**

The authors follow the trend of a multi-scale network game model and propose a framework to learn the utility functions of binary multi-scale games from agents' behavioral data. As the synthetic experiments shown (i.e. Figure 2), the method with group achieved much better results than the method without group.

Distinguished from previous work in this area, the authors model the agent behavior as logit-response dynamics, rather than a Nash equilibrium. After that, the authors formulate the game learning problem as MLE and prove that in the special case, i.e. with parametric multi-scale linear-quadratic utility models, the MLE problem is convex and can be solved efficiently. Finally, a likelihood ratio test was proposed which made it possible to statistically determine whether behavioral data generated by a multi-scale game model actually reflects multi-scale structure.

Generally speaking, the paper is well organized.

**Q4 Main Weakness:**

(1) It will be better to highlight the originality of this paper. I can understand the theoritical contributions of this paper, but it will be better to explicitly emphasize which ones are original and distinguish from the current SOTAs.
(2) It will be better to compare with the current SOTAs on synthetic datasets. And if the multi-scale network is the main contribution of this method, it will better to provide at least one SOTA also supporting the multi-scale network.
(3) The presentation of this paper can be improved.


**Q5 Detailed Comments To The Authors:**

(1) The title of introduction section: 2. Introduction -> 1. Introduction, and correspond changes should be made to the following sections.
(2) In the section of learning framework, it will be better to explain the notation when a Cartesian product first appears.

**Q7 Justification For Your Score:**

Generally speaking, the paper is well organized. However, the originality of this paper can be emphasized, the experiments can be more sufficient, and the presentation of the paper can be improved.

**Q9 Complying With Reviewing Instructions:**

1: Yes.

---

### Official Review · Reviewer_kjbd · 2022-04-19

**Q2(1) Originality/Novelty:** 3
**Q2(2) Significance/Impact:** 3
**Q2(3) Correctness/Technical Quality:** 3
**Q2(6) Clarity Of Writing:** 3
**Q6 Overall Score:** 6
**Q8 Confidence In Your Score:** 2

**Q1 Summary And Contributions:**

This paper proposes a game-theoretic generative model of time-series behavior data by combining single-shot multi-scale network games with logit-response dynamics. They model agent behavior as logit-response dynamics. Then this paper presents a general learning framework based on maximum likelihood estimation for inferring parameters of such games. The experiments show that the proposed modeling and learning approach is effective.


**Q2 Assessment Of The Paper:**

More detailed information regarding each of these aspects is given below:

**Q2(4) Quality Of Experiments (Optional):**

3: Good: The experimental evaluation is adequate, and the results convincingly support the main claims.

**Q2(5) Reproducibility:**

3: Good: Key resources (e.g., proofs, code, data) are available and key details (e.g., proofs, experimental setup) are sufficiently well-described for competent researchers to confidently reproduce the main results.

**Q3 Main Strengths:**

The paper presents its ideas in a clear and solid way.
The ideas are somewhat novel. The formulations are clean and neat .

**Q4 Main Weakness:**

Minor things. Some of the paper's figures can be resized and further polished.

**Q5 Detailed Comments To The Authors:**

1.The idea that defines a generative time-series model of joint behavior of both agents and groups is interesting and novel.

2.The Model and the Learning Framework Sections are well designed and well written. The technical parts in these two sections are very detailed and have sufficient mathematical derivation.

3.The experiments in this paper are complete, considering both synthetic data and real-world datasets. The experiment setting is quite rigorous, considering many situations, which is very convincing.

4.The experiments on real dataset are impressive. The preprocessing of the gang-related homicides dataset is effective and the visualization of the gang-related homicides dataset in supplemental material is intuitive and easy to understand. The experiments on bilateral trading data further demonstrates the promising application prospects of the proposed method in different fields.

5. I would suggest the authors to add one paragraph mainly talking about their contributions.

6.The size of the figures could be appropriately increased to help the readers better reading.

7.The x-axes of Figure 1 and Figure 2 should be labeled (length ).

8.Figure 3 mentioned five models, but we can only find curves corresponding to b-MSGN, Markov and LIG. The predictive log-likelihood trajectories on test data of Hawkes and Poisson are missing.

**Q7 Justification For Your Score:**

Minor things. Some of the paper's figures can be resized and further polished.

**Q9 Complying With Reviewing Instructions:**

1: Yes.

---

### Decision · Program_Chairs · 2022-05-15

**Decision:**

Accept (Poster)

**Comment:**

Meta Review: Overall, reviewers found that the paper advanced the state of the art in the problem by considering group-level reasoning. Interesting experimental results based on real data were also presented.